# Blood–Brain Barrier Disruption and Its Involvement in Neurodevelopmental and Neurodegenerative Disorders

**DOI:** 10.3390/ijms232315271

**Published:** 2022-12-03

**Authors:** Ana Aragón-González, Pamela J. Shaw, Laura Ferraiuolo

**Affiliations:** 1Sheffield Institute for Translational Neuroscience, University of Sheffield, SITraN, 385a Glossop Road, Sheffield S10 2HQ, UK; 2Facultad de Medicina, Universidad de Málaga, 29010 Málaga, Spain

**Keywords:** blood–brain barrier, neurodevelopment, neurodegeneration, therapies

## Abstract

The blood–brain barrier (BBB) is a highly specialized and dynamic compartment which regulates the uptake of molecules and solutes from the blood. The relevance of the maintenance of a healthy BBB underpinning disease prevention as well as the main pathomechanisms affecting BBB function will be detailed in this review. Barrier disruption is a common aspect in both neurodegenerative diseases, such as amyotrophic lateral sclerosis, and neurodevelopmental diseases, including autism spectrum disorders. Throughout this review, conditions altering the BBB during the earliest and latest stages of life will be discussed, revealing common factors involved. Due to the barrier’s role in protecting the brain from exogenous components and xenobiotics, drug delivery across the BBB is challenging. Potential therapies based on the BBB properties as molecular Trojan horses, among others, will be reviewed, as well as innovative treatments such as stem cell therapies. Additionally, due to the microbiome influence on the normal function of the brain, microflora modulation strategies will be discussed. Finally, future research directions are highlighted to address the current gaps in the literature, emphasizing the idea that common therapies for both neurodevelopmental and neurodegenerative pathologies exist.

## 1. Introduction

The blood–brain barrier (BBB) is a highly specialized and dynamic membrane formed by brain microvascular endothelial cells (BMVECs), astrocyte end-feet unsheathing the capillary, and pericytes embedded in the capillary basement membrane (BM), forming a functional element: the neurovascular unit (NVU) [1]. Due to its lipophilic nature, hydrophobic compounds and gases can diffuse across the BBB, but larger and hydrophilic compounds require specific transporters located within the barrier. A healthy BBB protects the brain from exogenous compounds and xenobiotics, filtering the blood flow [2]. Interestingly, there are BBB-absent regions in the central nervous system (CNS), including the circumventricular organs, the choroid plexus and the dura mater, allowing direct communication between them and blood [3,4,5,6]. These regions do not provide open circulation to the rest of the brain due to the presence of diffusion barriers such as the zonula occludens 1 (ZO-1) and glial fibrillary acidic protein (GFAP)-positive columnar cells between the area postrema (included in the circumventricular organs) and the nucleus tractus solitarius, which sends projections to the area postrema located in the brainstem [7]. 

As for brain formation and function, the neurodevelopmental period is a critical phase for the development of the BBB. This process will be briefly described to highlight the most important factors involved in each developmental step whose dysregulation may be implicated in various disorders [8]. 

BMVECs from the NVU are highly specialized cells and the major component of the barrier. They express important transporters such as ATP binding cassette (ABC) and synthesize relevant neurotransmitter molecules such as nitric oxide (NO), both of which are involved in neurodegenerative processes when dysregulated. Astrocytes, the most common cells in the brain, also have a relevant role in the BBB, secreting cytokines and exacerbating mechanisms contributing to neuroinflammation, a key player in both neurodevelopmental and neurodegenerative pathologies. In this review, the role of pericytes forming the barrier will be also discussed due to their part in the maintenance of BBB permeability. Moreover, basement membranes present in the BBB will be also described, offering to the reader a complete overview of the BBB anatomical structure [9].

To better understand the mechanisms by which the BBB could be altered, we will discuss the principal and most pathophysiological factors involved, including oxidative stress, hyperpermeability and, interestingly, the microbiota, dysfunction of which had been found in the context of neurodegenerative and neurodevelopmental disorders. Specific diseases such as amyotrophic lateral sclerosis (ALS), Alzheimer’s disease (AD) and autism spectrum disorders (ASD) will be taken as examples to highlight concepts and failures in the BBB system [10]. 

Efforts to facilitate access of potential therapeutic agents across the BBB will be highlighted. Recent findings connecting the microbiome and CNS disorders open the pathway to microbiology-based therapies. Other methods such as molecular Trojan horses or nanotechnology-based approaches which use the barrier properties to achieve brain penetration will be also discussed. Finally, BBB breakdown in pathology will be examined as an opportunity for stem cell-based therapies.

The main objective of this review article is to highlight the common mechanisms between neurodevelopmental and neurodegenerative disorders, revealing their similarities and proposing the use of analogous therapies in both types of disorder.

## 2. Development of the Blood–Brain Barrier (BBB)

“Barriergenesis” comprises a multiple-phase process: angiogenesis, differentiation, and maturation. These phases overlap spatially, at the cell level, and physiologically, at the molecular level. 

During **angiogenesis** (E9–E10.5 in mice and embryonic week (EW)-8 in humans [11]), endothelial progenitor cells (expressing the foetal liver kinase 1 receptor) from the perineural plexus invade the embryonic neuroectoderm following the concentration gradient of vascular endothelial growth factor (VEGF) and give rise to immature vessels [12]. VEGF is considered the major factor that controls brain angiogenesis. It is produced by cells in the subventricular neuroectoderm and reduced or absent neural VEGF results in impaired vascularization of the developing brain [13]. 

During this same period, the Wnt-β-catenin pathway plays an important role in the patterning and formation of the CNS. Neural progenitors express Wnt in the developing forebrain, the ventral regions of the neural tube, the dorsal spinal cord and the hindbrain. In particular, Wnt-7a and Wnt-7b have the broadest expression pattern in ventral regions of the developing CNS, and, in fact, Wnt-7b knock-out mice die by E11.5 [14]. 

Wnt ligands secreted by neural progenitor cells bind to frizzled receptors (FZD) expressed by endothelial progenitor cells responding to the VEGF gradient, leading to inhibition of β-catenin degradation via the proteasome. Subsequently, β-catenin accumulates in the cytoplasm and nucleus, where it induces transcription of target genes by interaction with transcription factors such as lymphoid enhancer-binding factor 1/T cell-specific transcription factor (LEF/TCF).

Wnt signalling activation leads to the transcription of BBB-related genes including those encoding glucose transporter 1 (Glut-1) and tight junction (TJ) molecules [15], such as *Cldn1* [16] and *Cldn3* [17] (Claudin). Hence, β-catenin is required for vessel formation as both the transducer of Wnt signalling and as a component of the adherent junctions (AJ) that join all the endothelial cells.

Some types of cells involved in BBB development, such as astrocytes, do not appear in a mature form until after birth. During the developmental phase, however, neural stem cells (NSCs), such as the radial glia, secrete transforming growth factor-β (TGF-β1) which plays a key role as a mediator of the interactions between glia and endothelial cells, contributing to the formation of the first blood vessels within the brain [18]. The interactions between these cells during this early phase give rise to the BBB, which in rodents is functional nearly as soon as it is established (E11 in mice), while TJs appear later in human development, at EW-14 [19].

Recent observations support the concept of ongoing barriergenesis in brain endothelial cells during angiogenesis [20,21]. Thus, barrier maturation in brain endothelial cells might not take place in two sequential phases as originally suggested [22].

Following BBB formation, the **differentiation** process (E15.5–E18.5 in mice) starts with pericytes and radial glia promoting barrier properties in the endothelial cells. Pericytes express the platelet-derived growth factor receptor β (PDGFR-β) at the endothelial surface, which acts as a signal for endothelial cells to be guided to the nascent vessel [23]. Pericyte recruitment to the developing endothelial capillaries is critical for the formation and maintenance of the BBB; in fact, PDGFR-β-deficient mice, which lack brain pericytes, die as a consequence of brain microhemorrhages [24]. Endothelial cell and pericyte interactions are mediated by bidirectional TGF-β–TGF-βR signalling. This signalling cascade leads to two crucial events: first, upregulation of endothelial N-cadherin, which promotes adhesion between pericytes and endothelial cells; second, pericytes are stimulated to deposit extracellular matrix components, such as angiopoietin-1 (Ang-1), contributing to basement membrane formation [25]. Moreover, Notch and sphingosine-1-phosphate signalling also contribute to the regulation of N-cadherin expression in brain endothelial cells [25,26]. The deficiency of PDGF-β or PDGFR-β leads to erroneous TJ distribution and increased vascular permeability [27]. As previously mentioned, radial glia contribute to the induction of barrier properties in endothelial cells, expressing SRC-suppressed C-kinase substrate (SSeCKS) which decreases the expression of VEGF through transcription factor AP-1 (activator protein-1) reduction and stimulates the production of Ang-1 [28]. Ang-1 is essential for normal vascular development; it binds and activates the Tek/Tie-2 receptor by inducing its dimerization and tyrosine phosphorylation [29]. Additionally, Ang-1 enhances the TJ formation, limits BBB permeability and reduces the expression of leukocyte adhesion molecules. The radial glia are also involved in limiting BBB permeability, releasing sonic hedgehog protein (Shh) which activates hedgehog signalling in endothelial cells through the receptor Patched-1 (Ptc-1) [30] and upregulates TJ proteins (ZO-1, occludin, and cldn-5) [31]. 

Finally, the **maturation** and maintenance of the barrier is the last phase in BBB development. It takes place postnatally and the timing is species-dependent [32,33]. Maturation is accomplished through TJ protein expression and their redistribution within the barrier. In this phase, radial glia differentiate into mature astrocytes. Wnt signalling and astrocytes regulate TJ formation because of the FZD receptor expressed by the endothelial cells. BBB formation and maintenance are supported by TGF-β–TGF-βR and Ang-1–Tie-2 signalling through pericytes and endothelial cells. Retinoic acid (RA) also contributes to BBB maturation, enhancing expression of AJ and TJ proteins [34]. Maintenance of BBB integrity is supported by astrocytes, which also produce apolipoprotein E (ApoE). It was shown that ApoE knock-out mice present with a dysfunctional BBB and develop psychotic behaviour, suggesting a relationship between impaired BBB function and neuropsychiatric diseases [35]. Interestingly, ApoE4 is a major genetic risk factor for development of Alzheimer’s disease (AD) [36]. In addition, astrocytes are the main cell type involved in the maintenance of the barrier, secreting Shh and Wnt in order to sustain BBB functionality throughout life [37].

## 3. Components of the Blood–Brain Barrier (BBB)

The neurovascular unit (NVU) has been described as a structure formed by microvascular endothelium, astrocytes, pericytes and neurons that are in physical proximity to the endothelium, basal lamina and parenchymal basement membrane. Each NVU component is intimately and reciprocally linked to each other, sharing several characteristics and establishing an anatomical and functional whole, which results in a highly efficient system regulating cerebral blood flow [38]. In this section, we will explore the main characteristics and functions of the NVU components.

### 3.1. Brain Microvascular Endothelial Cells (BMVECs)

BMVECs are a major cellular element of the BBB. They are extremely thin cells and present unique characteristics that distinguish endothelial cells of the brain from the vascular endothelium in the rest of the body, including tight junctions (TJs), absence of fenestrations, fewer or absent pinocytotic vesicles, expression of specialized transporters, and close association with other cell types comprising the NVU. These attributes allow them to tightly regulate the movement of ions, molecules, and cells between the blood and the brain [39]. 

The TJs are mainly composed of proteins such as ZO-1, -2,- 3, occludins and claudins; adherens junctions (AJs), including cadherins, actinin and catenins, and junctional adhesion molecules, e.g., JAM-1. ZO-1 is located on the cytoplasmic side of the BMVEC plasma membranes and connects the TJs with the cytoskeleton. JAM-1, on the other hand, participates in TJs formation in conjunction with occludin and claudin and is involved in cell-to-cell adhesion. In addition, JAM-1 is involved in leukocyte migration. Therefore, its dysregulation has been associated with alterations of CNS immunity [40]. Overall, dysfunction of the molecular components of BMVECs are widely associated with increased BBB permeability, thus exacerbating some pathological mechanisms such as oxidative stress, neuroinflammation, stroke or trauma [41] that will be further discussed later in this review (see Section 4). Additionally, BMVECs express two main types of transporters: efflux transporters [42] and highly specific nutrient/waste transporters [43]. Another remarkable difference between BMVECs and other endothelial cells is their higher content of mitochondria, which generate high levels of ATP used during active transport of ions and fluid [44,45]. 

Moreover, the quantity of leukocyte adhesion molecules is much lower in BMVECs compared to other endothelial cells, as these are involved in the interactions with leukocytes to regulate their transendothelial migration. In fact, the extremely low level of leukocyte adhesion molecules expressed in BMVECS is directly linked with the inability of immune cells to cross the barrier and enter into the CNS [40]. 

Interestingly, several biochemical studies have revealed the functional polarity present in the BMVECs, with different expression of enzymes, transporters, receptors and ion channels in the luminal and abluminal membrane surfaces of these endothelial cells to preserve brain homeostasis by controlling the exchanges between the blood and brain compartments [46].

Differences in enzymatic activity have also been found in BMVECs compared with other endothelial cells, with a high concentration of enzymes such as γ-glutamyl transpeptidase, alkaline phosphatase and aromatic acid decarboxylase. These enzymes take part in the assimilation of the neuroactive solutes originating from the blood, thus allowing BMVECs to metabolize drugs and nutrients for presentation to the brain [47]. 

#### BMVECs in Brain Vascular Contraction

BMVECs synthetize and release both endothelin-1 (ET-1) and NO which are in balance under healthy circumstances to maintain the function of the vascular tone. The disequilibrium of these molecules is involved in cerebral blood vessel dysfunction, such as in stroke [48]. ET-1 is one of the most potent vasoconstrictors known for mammalian blood vessels, with a relatively low concentration in plasma (0.2–5 pg/mL), while increased amounts have been reported in diseases such as hypertension and diabetes type 2 [49]. Endothelin receptors have been identified on platelets and blood vessels. Two subtypes of ET-1 receptors, ET_A_ and ET_B_, have been described; however, the predominant subtype in the brain is B [50]. ET_B_ opposes vasoconstriction by stimulating NO formation, acting as a feedback mechanism to limit the vasoconstrictor action of ET-1. NO inhibits platelet aggregation, the expression of adhesion molecules and the production of ET-1 [51]. The vasoconstriction mechanism through which ET-1 decreases the local brain flow is through platelet interaction, whereas a significant increase of ET-1 expression has been linked to haemorrhages [52]. Many physiological processes, including neurotransmission, are promoted by NO, which is mostly synthesized through endothelial NO synthase [53]. Interestingly, it has been reported that endothelial NO synthase knock-out mice have increased levels of amyloid-beta protein (Aβ) precursor, while expression of endothelial NO synthase and excess of NO have been associated with BBB disruption [54]. These data highlight the link between NO dysfunction, BBB disruption and some neurodegenerative disorders such as Alzheimer’s disease (AD) [55]. 

### 3.2. Astrocytes

Astrocytes are the most abundant cells in the brain and play important roles in the establishment and maintenance of the BBB. They occupy a strategic position between capillaries and neurons. Astrocytic end-feet form a coating network around the brain vasculature, the glia limitans, and, together with endothelial cells and pericytes, they form the BBB, separating the bloodstream from the brain parenchyma. Astrocyte dysregulation is associated with neurodegenerative diseases such as amyotrophic lateral sclerosis (ALS) [56] and AD [57] and paediatric neurological disorders such as Rett syndrome [58].

As mentioned in the previous section, astrocytes are in their immature form during BBB development; mature astrocytes, in fact, are not detected in the human foetal brain stem until the 15th week [59] and in foetal cortex until the 30th [60]. This late development, which continues even postnatally, offers a potential therapeutic window to reverse developmental dysregulation, thus making astrocytes an appealing therapeutic target. 

Although the heterogeneity of the astrocyte population across the brain is well known, it is still poorly characterized and little is understood about its impact on BBB function. In the mature barrier, astrocytes secrete cytokines, growth factors and extracellular matrix proteins through their end-feet, including, in particular, proteoglycans of the lectican family and tenascins [61]. Other aspects of glial support include the signalling of reinforcing pathways such as sonic hedgehog protein (Shh), vascular endothelial growth factor (VEGF), angiopoietin-1 (Ang-1), retinoic acid (RA) [62]. The role of astrocyte-derived Shh was proposed in maintaining the BBB via a mechanism involving regulation of TJ protein expression by BVMECs [37]. Nevertheless, transcriptomic studies have begun to point out that post-mitotic astrocytes, and not BMECs, represent the primary responders to hedgehog signalling in the adult brain [63]. Astrocytes are also involved in strengthening the TJ, whose maturation is promoted by release of glial cell-derived neurotrophic factor (GDNF), an EC ligand for the GDNF family receptor alpha-1 (GFRA-1). 

Furthermore, astrocytes play a crucial role in glutamate homeostasis, which is critical to maintain neuronal function and protect against excitotoxicity. Glutamate is known as the most abundant excitatory neurotransmitter in the mammalian nervous system, but its signalling is also important for the correct functioning of the BBB. Glutamate has been demonstrated to increase the permeability of BMVECs via activation of N-methyl-D-aspartate (NMDA) receptors [64,65]. For this reason, its clinical potential as a modulator of BBB permeability has been extensively explored in the context of neuroprotection and drug delivery [66]. Glutamate is also involved in blood flow regulation, since glutamate-mediated signalling prompts the release of NO from neurons, thus promoting vasodilation, and of arachidonic acid from astrocytes, which can have the double action of vasodilator or vasoconstrictor [67].

Considering the pivotal role of glutamate in the CNS and the injurious effects of its excessive accumulation in the synaptic cleft, astrocytes express excitatory amino acid transporters (EAATs), i.e., Na^+^-dependent glutamate transporters, in order to keep the concentration of glutamate tightly controlled in the extracellular space [68]. In addition, astrocytes are also involved in BM regulation by using ammonia in the synthesis of glutamine, metabolizing short-chain fatty acids, taking part in the regulation of brain nitrogen metabolism, and, finally, preventing the accumulation of ammonia, glutamine and glutamate in the CNS [69]. 

Another critical role of astrocytes in maintaining BBB function is the release of TGF-β, which regulates multiple biological processes, adult stem cell differentiation, immune regulation, apoptosis, and inflammation [70]. In addition to the pivotal role of TGF-β in TJ and blood vessel formation described above, TGF-β also downregulates endothelial anticoagulant factors, such as thrombomodulin, and increases blood flow under pathological conditions [71]. 

Similarly involved in fluid exchange and regulation, aquaporin-4 (AQP4) is the most abundantly expressed water channel in the brain and is predominantly expressed in the end-feet of astrocytes [72]. AQP4 regulates water permeability and plays an important role in neuroimmunological functions too, but its role in brain physiology has remained elusive for years. AQP4 controls bidirectional fluid exchange [73] and has been linked to several pathological processes including paediatric brain neoplasms with dysfunctional BBB [74] and the neuroimmunological disorder neuromyelitis optica [75]. 

### 3.3. Pericytes

Pericytes coexist with the astrocytes in the abluminal compartment to maintain the BBB properties [76], located within the NVU between endothelial cells, astrocytes, and neurons. The relevant role of pericytes in the developmental phase of the barrier and their interaction with endothelial cells through PDGF-β signalling during the process has been described above. 

It was shown that the number of pericytes involved in the barrier inversely correlates with its permeability, thus a decrease in the number of pericytes correlates with an increase in the BBB permeability [77]. In addition, reduction in pericyte coverage across the BBB is inversely correlated with ageing [78] and neurodegeneration, as in ALS [79].

In the case of pericyte loss, as it occurs in ageing, brain injury or neurodegeneration, these cells can actively adapt to ensure endothelial coverage by extending their wide-reaching processes [80]. Pericytes are major regulators of cerebral blood flow due to their sensitivity to glutamate signalling, which promotes the release of vasodilators such as NO and prostaglandin E_2_ [81]. They also help direct astrocyte foot projections and are responsible for reducing levels of leukocyte adhesion molecules [82]. 

Additionally, pericytes control the expression of TJ and AJ proteins and their alignment. They also regulate transendothelial vesicle trafficking across the barrier which is implicated in the transport of nutrients and essential molecules [82]. These cells also play a major role in blocking the entrance of xenobiotics, including therapeutic compounds, into the brain, which makes them a potential target for drug delivery [83]. 

The role of pericytes in neuroinflammation has been demonstrated by Olson et al. [84] in conditional PDGFR-β knock-in mice. These researchers showed the way in which a PDGFR-β-induced immune response could modulate the inflammatory properties of endothelial cells, leading to increased leukocyte adhesion and transmigration. They also revealed that amplified PDGFR-β signalling led to higher pericyte coverage of blood vessels and induced changes in pericyte differentiation. On the other hand, pericyte degeneration results in BBB breakdown with the accumulation of neurotoxic molecules leaking from the blood [82]. 

### 3.4. Basement Membrane (BM)

The BM is the non-cellular component of the barrier and it is a unique form of extracellular matrix. Its main functions are structural support, cell anchoring and signal transduction [85]. There are two types of BMs separating endothelium from astrocytes. One of the membranes is composed of fibronectin, collagen type IV, nidogen, perlecan and laminin and it is denominated as the endothelial BM. The other one is the perivascular glia limitans, also known as astroglial or parenchymal BM and is formed by fibronectin, agrin and laminins [86]. Consequently, the most important biochemical components of the BM are collagen type IV, fibronectin and laminin.

Collagen-IV and fibronectin are secreted by the cellular components of the NVU. **Collagen-IV**, the most abundant component of the BM, maintains BM stability by retaining other protein components such as laminin, perlecan and nidogen. Six collagen-IV isoforms have been identified of which collagen-IV α-1/2 are present in almost all BMs and are highly conserved across species [87]. Nonetheless, there is evidence that collagen-IV is not necessary for early embryonic development, but it is indispensable for the structural integrity of the BMs at later stages [88]. To exemplify the importance of collagen-IV α-1 in BMVECs and astrocytes, mutations affecting the coding gene contribute to cerebrovascular defects resulting in intracerebral haemorrhages [89]. 

On the other hand, **fibronectin** stimulates the proliferation and survival of the endothelial cells in the BBB [90] and, in fact, both fibronectin and collagen-IV knock-out mice are embryonically lethal. Defects in the mesoderm, impaired neural tube and vascular development are caused by the absence of fibronectin [91], while collagen-IV deficient mice display structural deficiencies in the BMs with impaired integrity of Reichert’s membrane, a basement heath between the parietal endoderm cells and trophoblast cells.

BMVECs, pericytes and astrocytes synthetize different **laminin** isoforms. There is a cell-specific expression pattern, hence laminin shows differential distribution between endothelial and parenchymal BMs. As an example, astrocytes produce different laminin isoforms depending on their BM location. Hence, laminin-211 is most abundant in the parenchymal membrane and laminins-411/511 are predominantly expressed in the endothelial membrane [92]. Deletion of laminin full isoforms is lethal during embryonic development. Similarly to fibronectin mutant mice, laminin-211 deletion leads to intracerebral haemorrhage and also age-dependent BBB breakdown [93].

To analyse the significance of laminin in the regulation of the BBB, Menezes et al. [94] generated mice lacking expression of the laminin α2 subunit within the laminin-211 heterotrimer expressed by astrocytes and pericytes. They reported altered integrity and composition of the endothelial basal lamina, inappropriate expression of embryonic vascular endothelial protein MECA-32, reduced pericyte coverage, and TJ abnormalities. Their data reveal the role of laminin in regulating the interactions of the NVU cellular components within the BBB. 

The effect that lack of laminin chains has on the BM and BBB integrity was explored in several studies, but the mechanisms driving these phenomena are still largely unknown (see Yao et al. for a detailed review [95]). 

### 3.5. Transport across the BBB

The BBB is lipophilic in nature, hence hydrophobic molecules and gases such as CO_2_ or O_2_ can cross it by passive diffusion [96]. However, only solutes of a molecular weight below 400 Daltons (Da) are able to circulate freely through the BBB endothelium [97]. Because it isolates the brain from the rest of the body, many polar nutrients are needed but cannot diffuse across the barrier. Thus, the main factors that influence the ability of circulating molecules to cross the BBB are their polarity and their size.

Additionally, the TJs forming the BBB act as a barrier to segregate transporters to the abluminal or luminal membrane face, thus preventing their movement across the endothelium and maintaining BBB polarity. Some transporters are present on both sides of the membrane or just in one of them, depending on the brain requirements for nutrients and the region [76]. 

In addition to passive transport, multiple molecules can be shuttled across the BBB through a variety of ion channels and selective transporters. The main transport mechanisms can be divided into endothelial cell and pericytal transport with machineries across both cell types including active efflux [98], carrier-mediated (CMT [99]) ion-transport [100] and receptor-mediated transport (RMT) [101], with exception of active efflux, which is a specific property of BMVECs. In addition to these, there is also a BMVEC/pericyte independent mechanism, vascular-mediated transport [102]. 

As mentioned before, BMVECs express two main categories of transporters: efflux transporters (i.e., ABC- and EEAT-transporters), which transport lipophilic compounds, and nutrient transporters, regulating the exchange of nutrients and removal of waste products. For example, essential nutrients such as glucose or amino acids are transported across the barrier by specific solute carriers to supply the essential substrates for brain metabolism [103]. 

These transport mechanisms and their functional relevance in health, together with their role in many disorders such as AD [104], ALS [105], Huntington’s disease [106], schizophrenia [107], Parkinson’s disease [108] and microcephaly [109], among others, have been extensively described previously [110]. 

## 4. Mechanisms Altering Blood–Brain Barrier (BBB) Function

The BBB is a complex system where cells and structural proteins create an environment in which any alteration can affect the normal activity of the others. For that reason, stress mechanisms such as oxidation, mechanical insults or genetic factors may result in an imbalance of the BBB. Several elements affecting the neurovascular unit components have been described elsewhere [111], but in this review we will focus on those most relevant for their impact on neurodegenerative and neurodevelopmental diseases (Figure 1).

### 4.1. Genetic Factors

#### 4.1.1. APOE

*APOE* (chromosome 19) encodes apolipoprotein E which binds to a specific liver and peripheral cell receptor and is essential for the normal catabolism of triglyceride-rich lipoprotein constituents [112]. In the CNS, APOE is expressed mainly in astrocytes [113], but also microglia, vascular mural cells and choroid plexus cells. APOE modulates multiple pathways that affect cognition, thus involving lipid and glucose metabolism, synaptic function, neurogenesis and neuronal degeneration, neuroinflammation, mitochondrial function, tau phosphorylation, and Aβ metabolism. Moreover, APOE has a crucial role in amyloid beta-protein (Aβ) clearance, aggregation and deposition [114].

The ε4 allele of *APOE* (*APOE4*) constitutes the major susceptibility gene for late-onset Alzheimer’s disease (AD) [36], and, in fact, it has been associated with both decreased neuroprotection and increased neurotoxicity [115].

In the context of the BBB, APOE4 affects barrier function by activating the pro-inflammatory protein cyclophilin A via the nuclear factor-kB (NFkB)–matrix-metalloproteinase-9 (MMP-9) or NF-kB-MMP-9 pathway in pericytes. Studies with transgenic *APOE2* and *APOE3* mice have indicated that ApoE2 and ApoE3 maintain the BBB structure by suppressing the NF-kB-MMP-9 pathway through activation of LRP1. On the contrary, ApoE4 fails to activate this protective mechanism, thus triggering tight junction (TJ) and basement membrane (BM) degradation, and therefore BBB breakdown [116]. This is supported by post-mortem human frontal cortex tissue samples where the pericytes of APOE4 carriers showed loss of integrity of the BBB [117].

That loss of typical BBB function is an important upstream event in AD and is supported by the fact that, although atrophy of the hippocampus is considered an early biomarker of disease [118], more recent studies demonstrate that BBB breakdown occurs even before atrophy [119]. This early BBB breakdown might appear as cerebral microbleeds, which are frequently seen in AD patients, particularly in *APOE4* carriers, which involve severe BBB breakdown [120]. This suggests that BBB failure might precede neurodegeneration. Interestingly, *APOE3* carriers have a reduced rate of BBB failure, in addition to a reduced risk for the development of AD [121].

The relevance of APOE is not only studied in AD progression, but it has also been related to ageing and neurodegeneration and in pathologies such as vascular dementia, Parkinson’s disease (PD) or ischaemic stroke, where APOE seems to participate in the progression of these diseases, although the underlying mechanisms of action are still unclear [122].

In conclusion, more research is needed to clarify not only the role of *APOE4* in neurodegeneration and ageing, but also in overall brain functioning.

#### 4.1.2. SOD1

*SOD1* (chromosome 21) encodes for the enzyme copper/zinc superoxide dismutase 1. SOD1 is widely expressed throughout the human body, particularly in liver, because of its detoxifying function, and in the CNS, probably due to the high metabolic rate in the region. SOD1 is located predominantly in the cytoplasm, but some pathogenic variants gradually aggregate and accumulate in mitochondria [123]. Mutations in *SOD1* can cause conformational instability or misfolding of the SOD1 protein, and comprise the second major genetic risk factor associated with amyotrophic lateral sclerosis (ALS), accounting for 20% of familial ALS (fALS) cases [124].

Although rodent models have clarified that in general SOD1 mutations are not causative of ALS through a loss of function, some specific mutations result in decrease or loss of enzymatic activity and thus insufficient degradation of ROS [125]. The consensus is that SOD1 mutations cause ALS through a complex toxic gain of function which leads to an increase in oxidative stress, protein misfolding and aberrant protein interactions [126].

In terms of BBB function, mutant *SOD1* rodent models develop a leaky BBB with higher permeability, enlarged astrocytic end-feet, interrupted BMs associated with a reduction of BMVECs and astrocytes, thus leading to oedema and microbleeds. This pathological phenotype is also observed in ALS patients [127].

SOD1 dysfunction has been linked not only with ALS. Due to triplication of chromosome 21, this protein is highly overexpressed in brains of patients with Down syndrome resulting in increased production of H_2_O_2_ [128]. This affects mainly the immune response and the microbiome through modification of signalling pathways leading to activation of phagocytosis as well as an imbalance in the concentration of superoxide free radicals [129]. Hence, greater SOD1 activity may impair neutrophil chemotaxis, thus increasing the risk of bacterial infections in patients [130].

#### 4.1.3. AQP4

As previously mentioned, *AQP4* (chromosome 18) encodes a water-selective channel 4 in the plasma membrane, the major water channel within the brain, which is concentrated in astrocytic foot processes at the BBB [131]. Apart from water, AQP4 conducts aquaglyceroporins that mediate diffusion of water, glycerol, and other small molecules without charge across membranes [73]. This channel is considered responsible for brain water homeostasis and central plasma osmolarity regulation, with a possible effect on the activity of inward rectifier potassium channel 4.1 (KIR4.1). AQP4 and KIR4.1 are co-expressed and together compose a multifunctional unit participating in the clearance of water and K^+^ following neural activity [132]. Additionally, this water channel appears to participate in neurotransmission [133], synaptic plasticity [134], astroglial cell migration during glial scar formation, stroke and tumours or abscesses [135]. AQP4 is known to be altered in several neurological conditions, including tumours, neuromyelitis optica [136], autism spectrum disorders [137], AD, ALS, multiple sclerosis (MS), stroke, epilepsy and as a consequence of traumatic brain injury [138].

Moreover, AQP4 is also related to Aβ clearance in AD. To explore the role of this channel in AD pathology progression, an AD mouse model lacking Aqp4 was used by Smith et al. to show that Aqp4-deficient mice displayed increased amyloid deposition, impairment of peri-plaque astrocyte structural organization and recruitment of microglia to plaques, thus amplifying the damage inflicted upon neurons adjacent to the plaques [139].

Some studies have also associated AQP4/KIR4.1 alteration to neurodevelopmental disorders. As previously highlighted, AQP4/KIR4.1 are key modulators of brain homeostasis, which is particularly critical during neurodevelopment. Disruption of the AQP4/KIR4.1 balance is likely to facilitate brain oedema formation possibly resulting in an increased risk for sudden infant death syndrome. Indeed, Opdal et al. conclude in their research that *AQP4* may be a predisposing factor for this disorder [140].

In support of this hypothesis, a recent publication has explored the upregulation of AQP4 in a double knock-down mouse model with promising results when rescuing the channel activity by an enhancer. The authors suggest a pivotal role of AQP4 in suppressing oedema formation by maintaining BBB integrity and upregulating TJs [141].

### 4.2. Oxidative Stress

Oxidative stress is a phenomenon caused by an imbalance in the production and accumulation of reactive oxygen species (ROS) in cells and tissues and the ability of the biological system to detoxify them [142]. The most common ROS are superoxide anions (O_2_**^−^**), hydrogen peroxide (H_2_O_2_), hydroxyl radicals (·OH) and oxygen (O_2_). They are generated by physiological cell metabolism; in fact, the mitochondria are considered a primary site of ROS production from aerobic respiration. Increased oxidative stress has been extensively linked with different kinds of disorders such as cancer and diabetes or ALS and PD [143]. ROS sources can be also exogenous and can be found in pollutants, tobacco, xenobiotics, ultraviolet radiation, pesticides, etc. These particles can trigger the production of inflammatory cytokines, e.g., tumour necrosis factor-alpha (TNF-α) that operate through ROS-related mechanisms within the cell [144], involving neuroinflammation and oxidative stress.

In addition, ROS have been demonstrated to cause BBB dysfunction. One example is the way alcohol abuse can induce the activation of myosin light chain kinase, leading to phosphorylation of TJ proteins and enhanced monocyte migration across BBB, thus decreasing BBB integrity [145]. Similarly, ingestion of alcohol during pregnancy is associated with developmental diseases such as foetal alcohol syndrome. In this condition, neonates develop impairment of BBB function and neurodegeneration due to the increased production of ROS in the brain by catalase-mediated acetaldehyde production from alcohol [146].

#### 4.2.1. Antioxidant Defences

The role of the antioxidants in combating oxidative stress is vital for physiological brain function. The reduction of this natural defence is a common factor in multiple pathologies which has attracted a lot of interest especially with the demographic increase in neurodegenerative diseases within the population in recent years [147].

##### Enzymes

Antioxidant enzymes constitute one of the principal intracellular defence mechanisms. The **Nrf2** (nuclear factor erythroid 2-related factor 2) is a transcription factor constitutively expressed in all tissues [148]. Nrf2 plays a key and complex role in the response to oxidative stress, it is a master regulator of a large transcriptional response to cellular stress. Nrf2, in fact, recognises an enhancer sequence known as the antioxidant response element (ARE), which is upstream of several antioxidant and anti-stress genes [149]. In addition, as part of its transcriptional response, Nrf2 downregulates the expression of some TJ proteins such as occludin and cldn-18 [150].

The therapeutic potential of Nrf2 activation has been and is still being explored in multiple disorders, including PD, AD and ALS, where various disease models highlight its importance in the pathophysiology of the disease [151].

Another important enzyme family implicated in the antioxidant defences is the family of the SOD proteins. SOD1 is a cytoplasmatic protein that catalyses the dismutation of superoxide anions to hydrogen peroxide and, interestingly, mutations in this enzyme have been the first associated with ALS. On the contrary, the SOD2 isoform is located in the mitochondria and dismutates superoxide to the less reactive hydrogen peroxide [152]. SOD2 reduction, as in the Sod2^+/−^ mouse model, accelerates the onset of behavioural changes for AD. Interestingly, the reduction was associated with both a decrease in Aβ deposition in the brain parenchyma and an increase in amyloidosis in the brain vasculature [153]. Lastly, the isoform SOD3 is the extracellular anti-oxidant member of the family, which can be detected in fluids such as plasma or CSF. Furthermore, it has been shown that SOD3 synthesis by human bone marrow-derived mesenchymal stem cells is regulated synergistically in response to the inflammatory mediators TNF-α and interferon-gamma (IFN-γ). This induction in response to inflammation provides further evidence of the role of the SOD3 in preventing neuronal and axonal injury against NO or microglial-induced damage [154]. Additionally, SOD3 has been explored as a potential biomarker for neuroinflammation and neurodegeneration, since an increased level was reported in CSF from ALS patients [155].

Similarly, catalase is another important antioxidant enzyme which is ubiquitously expressed in tissues that utilise oxygen. By using an iron or manganese cofactor, it catalyses the reduction of hydrogen peroxide produced in the detoxifying cascade by SOD1 to form water and oxygen [156]. As previously mentioned, Shaw et al. demonstrated an increase of SOD3 in ALS along with the upregulation of catalase.

##### Non-Enzymatic Anti-Oxidants

**Glutathione** is a tripeptide containing cysteine, glycine and glutamic acid and its synthesis is carried out by two enzymes, glutamate cysteine ligase and glutathione synthetase. Glutathione can be found in two states, reduced (GSH) and oxidized (GSSG), and the ratio of GSH and GSSG is indicative of the redox status of the cell. In the brain, glutathione is considered the main antioxidant molecule, with an approximate concentration of 2 mM, much higher than in the CSF, where the concentration is only 4 μM, or in the plasma where it is around 2 μM [157]. Glutathione has critical roles in cellular antioxidant defence, including neutralizing oxygen, hydroxyl radicals, and superoxide radicals. In the brain, it also takes part in the detoxification of metals such as mercury, the regulation of cellular proliferation and apoptosis, mitochondrial function and the maintenance of its DNA, and it acts as a cofactor for signal transduction [158]. In the CNS, astrocytes have an elevated concentration of glutathione compared to neurons and they also secrete this molecule into the extracellular space for protection of neurons [159]. Huang et al. investigated the GSH shuttle from astrocytes to endothelial cells. GSH was found to preserve endothelial barrier stability by maintaining TJ proteins and preventing injury-induced TJ phosphorylation. In addition, supplementation with GSH analogues showed positive effects in supporting brain vascular stability and homeostasis [160].

Another important player in this regard is **vitamin C** or ascorbic acid, which is considered a vital antioxidant within the brain where it is more concentrated compared to blood levels. Besides the antioxidant role of ascorbic acid, this molecule is also implicated in other functions including neuronal maturation and differentiation, collagen maturation and modulation of neurotransmission. Vitamin C can cross the BBB in two ways, through the sodium-dependent vitamin C transporter (SVCT) or in its reduced form as dehydroascorbic acid (DHA) through the Glut family transporters [161]. Various studies have demonstrated the impact of vitamin C on neurodegenerative and neurodevelopmental disorders.

In particular, the importance of vitamin C during pregnancy to support normal foetal growth and development has been extensively explored [162]. For example, vitamin C deficiency has been linked to severe impairment of vascular development due to its relevance in angiogenesis, e.g., in collagen maturation [163]. Interestingly, a recent study reported that the parenteral administration of vitamin C significantly attenuates disruptions of BBB and protects TJs (i.e., Cldn-5 and ZO-1) in a rat model of cerebral ischaemia [164].

**Vitamin E** is also implicated in several cellular anti-oxidant defence mechanisms. In fact, it has been reported that a deficiency of vitamin E can trigger increased levels of oxidative stress, thus causing a disruption of the integrity of the BBB with important consequences for CNS function and maintenance [165]. Moreover, vitamin E supplementation seems to restore neuromuscular function in rodents and humans [166]. In this context, vitamin E deficiency has been linked with the pathogenesis of motor neuron diseases where oxidative stress plays a key role, including free radical damage to motor neurons in ALS [167]. In vivo studies assessing the effect of vitamin E treatment on disease progression in ALS mouse models are controversial, while some studies found a beneficial effect as vitamin E could reduce oxidative stress markers after 3 months of vitamin E supplementation with riluzole, even though no improvement of patient survival was reported [168]. Other studies have shown little or no effect and advised further investigation [169].

In summary, this evidence suggests a strong relationship between oxidative stress, BBB impairment and neuronal injury in various neurodegenerative and neurodevelopmental disorders.

### 4.3. Neuroinflammation

Neuroinflammation is a key pathophysiological factor in the exacerbation of neurodegeneration, but is also implicated in neurodevelopmental disorders and in BBB impairment [170]. The mechanisms by which neuroinflammation may disrupt the BBB are not fully understood. However, activation of microglia, astrocytes and endothelial cells leading to the secretion of pro-inflammatory cytokines such as TNFα and interleukin-1 beta (IL-1β) have been demonstrated as components of neuroinflammation [171]. In addition, increased expression of endothelial adhesion molecules, modifications of TJ proteins and upregulation of matrix metalloproteinases which increases BBB permeability by degrading TJs and extracellular matrix components of the endothelial BM have been described [172]. All these events lead to BBB leakiness, which in turn allows pathogen and immune cell invasion.

Immune cell infiltration and leakage of cytokines from the blood may aggravate brain damage after brain injury or under hypoxic conditions. Experiments using cromoglycate, a mastocyte stabilizer which inhibits cell degranulation, resulted in decreased BBB opening, reduced glial activation and neuronal death, allowing long-term neuroprotection [173]. Several studies have reported acute dysregulation of the immune system in ASD patients including chronic neuroinflammation. For example, elevated levels of pro-inflammatory cytokines such as IFN-γ, IL-1β, IL-6 and TNF-α and chemoattractant protein-1 (MCP-1) from macrophages accompanied by accumulation of microglia and astrocytes around the blood vessels have been reported in post-mortem ASD brain samples as reviewed by Bjorklund et al. [174]. Similar pathology has been described in several neurodegenerative diseases such as AD, PD or ALS, where activated microglia and reactive astrocytes exacerbate the inflammatory cascade and the disruption of the BBB [175].

However, the physiological role of inflammation cannot be overlooked, as clear evidence supports the idea that neuroinflammation may be favourable in the acute stage of traumatic brain injury to clear damage and set the stage for remodelling events [176].

### 4.4. Injures to the BBB

As highlighted earlier in this review, correct filtration of solutes, chemicals and cells through the BBB is essential for CNS homeostasis and physiological function, while several studies have implicated BBB disruption in neurodegenerative processes. One of the most common events reported in several neurodegenerative conditions is immune cell invasion from the blood stream, as observed in post-mortem CNS tissue from ALS patients. Studies have reported infiltration of mast cells and neutrophils along the motor pathways and at the neuromuscular junction, mediating degeneration [177]. Although some studies suggest that disruption of the BBB is an early event in ALS and others indicate that BBB hyperpermeability is involved in later disease stages, this process of BBB impairment has been associated with motor neuron degeneration [178]. Less debated is the involvement of microbleeds, with erythrocyte extravasation and brain infiltration by peripheral macrophages and neutrophils in AD post-mortem samples, which suggest that the brain’s innate immunity is activated; this may exacerbate the neurodegenerative process [179]. Moreover, cerebral microhaemorrhages or microbleeds are known to occur in cerebrovascular-linked neurodegeneration, dementia and ageing. Although microbleeds have been linked with neurodegeneration [180], it is still unclear whether they precede and cause neuronal damage or whether they are a result of the neurodegenerative process.

One of the main mechanisms causing BBB damage is **traumatic brain injury (TBI)**, which encompasses a complex pathogenic process that results from primary and secondary injuries, thus leading to temporary or permanent neurological deficits [181]. Consequently, these processes greatly alter the normal functional interactions between glial cells and the cerebrovascular endothelium, including the expression of transporters and TJs across the BBB and increased oxidative stress among other pathophysiological processes [182]. Disruption of the BBB is such an important component of TBI-related neurodegeneration that the BBB is has been proposed as a target for therapeutic intervention.

In contrast, **hypoxic-ischaemic encephalopathy (HIE)** is the major cause of brain injury in neonates, resulting from oxygen and/or blood flow deficiency during the gestational period. As in TBI, the brain’s response to global hypoxia-ischaemia is a multistep process. After an initial phase of cerebral flow alteration and vasoconstriction resulting in oedema [183], processes such as neuroinflammation, mitochondrial permeabilization and further oxidative stress take a central role [184]. Similarly to what happens in TBI, the hypoxic process causes disruption of TJ proteins and an increase in BBB permeability. These events are mediated by pro-inflammatory cytokines such as TNF-α and interleukins; common factors in neuroinflammation and also implicated in neurodegeneration [185].

Another gestational complication that compromises the BBB is **foetal intrauterine growth restriction (IUGR)**, affecting up to 10% of pregnancies and considered as the second most common cause of perinatal mortality. Most often it results from chronic hypoxia in utero caused by placental dysfunction, which entails a reduction in the transfer of oxygen and nutrients from the mother to the foetus, damaging the brain in several ways, with impairment of brain growth, vascular development and severe alterations of neurovascular unit organization [186]. Castillo-Melendez et al. reported a reduction in pericytic and astrocytic coverage and reduced BBB integrity in a lamb model of IUGR. These alterations may lead to cognitive impairment, defects in short term memory, speech delay, abnormal behaviour as attention deficit hyperactivity disorder or ASD, among others [187]. More long-term investigation is needed to better understand the implications in adult life, for example a higher predisposition to suffer from neurodegeneration for IUGR infants (see Kesavan et al. for a comprehensive review of IUGR and its complications [188]).

### 4.5. Gut–Brain Axis: Microbiota

During the last decade, increased interest in the importance of the gut microbiota in neuronal development and behaviour as well as neurodegeneration has created a “paradigm shift in neuroscience” by indicating that the gut may play a pathophysiological role in several human brain diseases, including ASD, anxiety, depression, and chronic pain [189,190]. The emerging literature has demonstrated bidirectional signalling between the brain and the gut microbiome involving multiple neurocrine and endocrine signalling mechanisms [191].

An obvious area of investigation are the molecules involved in the communication between the gut microbiome and the brain. Mayer et al. have carefully reviewed this aspect, highlighting the endocrine, neurocrine and inflammatory-related signals generated by the gut microbiota that can affect the brain of mice. In turn, the brain can influence the microbial composition and function in the gut via endocrine and neural mechanisms including sensing luminal metabolites by intestinal enteroendocrine cells and communicating to the brain via activation of the vagus nerve by hormones and neurotransmitters, and via neural routes such as neuroepithelial connections [191,192].

Moreover, microbiota have been proposed to be involved in BBB modulation in health and its dysregulation could contribute to BBB dysfunction in neurodegenerative disorders, e.g., PD [193], and neurodevelopmental disorders, e.g., ASD [194]. Recent studies have highlighted a relationship between ASD, inflammation and the gut microbiome. ASD patients have been reported to demonstrate microbiota alterations and gastrointestinal abnormalities. The disruption of intestinal barrier integrity is implicated in both immune and inflammatory responses. What is even more interesting, research conducted on animals showed that the composition of gut microbiota also influences the integrity of the BBB, with evidence that the gut microbiota affects the prenatal development of the BBB and its permeability later in life [195]. Hsiao et al. found that *Bacteroides fragilis* caused an increased expression of the pro-inflammatory cytokine IL-6 in the colon [196]. In relation to this, one could speculate that if pregnant women experienced infections during critical periods of pregnancy, this could trigger systemic inflammation and influence the neural development of the developing foetus.

Interestingly, there is also a new field in neurodegenerative research related to the **oral microbiome** [197]. The mechanism of action is similar to that previously described, where oral microbiota could be transported from the mouth to the brain through the bloodstream during brushing, flossing, chewing, or toothpick use in patients with periodontitis, causing bacteraemia [198]. In addition, the increase in pro-inflammatory responses can weaken the BBB, promoting the penetration of pathogens as previously mentioned. In this context, elevated levels of periodontal disease bacteria were found in AD patients. Recent studies have identified oral microbiota that produce Aβ, referred to as microbiome-derived amyloid. This can access the rest of the body and may be able to promote chronic CNS pathologies with amyloidogenic features such as AD [199]. Studies in rodents also suggest that infections leading to an increase of the brain inflammatory state during early life and adulthood can promote cognitive decline and neurodegeneration progression in later life. In addition, they demonstrated that pathogen-free conditions slow the onset of neurodegeneration in transgenic mice [200].

To conclude, the essential role of the microbiome in health and disease may provide new insights in the field; despite the fact that our knowledge of this complex brain–gut microbiome cross-talk is currently limited, there is research impetus to generate greater understanding of this system in health and disease.

## 5. Potential Therapies

The unique properties of the blood–brain barrier (BBB) not only confer cerebral protection from toxic insults, but also underpin a major challenge to drug delivery within the CNS. Several drugs have been developed to target CNS disorders but cannot cross the BBB. Therefore, great effort had made to design improved access to the CNS of potential therapeutic agents.

Even though the circumventricular organs were described by some researchers as “brain windows” due to the lack of BBB [3,4,5,6], it should be noted that in these regions the capillary endothelial cells display higher enzymatic activity which limits the entry of neurotransmitters, toxins and drugs, thus forming an “enzymatic barrier” [201].

As Dong et al. reviewed extensively, there are multiple strategies for brain drug delivery that can be summarized into three main classes: neurosurgical-based, chemistry-based and biology-based strategies. The first is an invasive method involving intracranial surgery, the others are based on increasing the lipid solubility of the drug or the reformulation of molecules to cross the barrier utilising endogenous transporters within the BBB [202]. Lately, stem cell-based techniques are being considered a powerful therapeutic resource for some disorders (Figure 2).

### 5.1. Neurosurgical-Based Techniques

With the administration of drugs by intracerebroventricular infusion, the compounds arrive at the ependymal surface and their concentration in the brain parenchyma decreases exponentially with the distance from the ependymal surface. On the other hand, with the delivery by intracerebral implantation, the drug distribution is restricted to the site of delivery and, consequently, the therapy may be restricted to a small region within the brain. Similar diffusion problems have been reported with the convection-enhanced diffusion technique, which aims to replace diffusion with convection. Nevertheless, these are promising techniques which are under investigation and which may be applied in the future to a variety of CNS disorders [203,204].

### 5.2. Chemistry-Based Techniques

With the increase in drug repurposing efforts for CNS disorders, the need to modify approved and effective molecules used for diseases affecting peripheral organs has also increased.

Another approach to increase drug delivery to the CNS is the chemical modification of small molecules in order to make them more BBB penetrant. To increase drug lipid solubility, various approaches can be employed, but the most common are the direct addition of lipophilic groups or halogenation. Halogenation is a chemical process consisting of the replacement of a hydrogen atom by a halogen atom as chloro or bromo, as shown by Gentri et al. [205]. Another chemical reaction often used is methylation; in fact, adding a methyl group can reduce the hydrogen bonding potential of peptides with enhancement of lipophilicity. Although there are successful examples of this process, chemical modifications leading to improvement in BBB permeability can also affect the properties of the original molecule, thus leading to potential decrease in specificity or potency. To illustrate the relevance of correct lipophilic mechanism and the intrinsic problems, Witt et al. explored the stereoselectivity differences in a synthetic opioid, DPDPE. They showed that the methyl group position on the phenol benzene ring can significantly modulate receptor selectivity, binding capacity, lipophilicity, or BBB penetration [206].

Lipophilicity has been shown to be a major determinant of drug diffusion across the BBB [207]. However, increasing the lipid solubility entails several problems such as drug-affinity reduction and increase in drug molecular weight, making BBB penetration potentially more difficult and resulting in higher non-specific tissue absorption [202]. 

### 5.3. Biology-Based Techniques

#### 5.3.1. Trojan Horse Technology

To overcome the issues related to other approaches, researchers considered exploiting the properties of the BBB itself by using endogenous transporters to cross the barrier. One example is the molecular Trojan Horse technology, which by genetic engineering allows the use of peptides and recombinant proteins, enzymes and other large molecule drugs to cross the BBB via a receptor mediated transport (RMT) process [208]. Neurotrophins such as brain-derived neurotrophic factor (BDNF) or basic fibroblast growth factor (bFGF) have been frequently explored since they are secreted proteins which are essential for neuronal development and survival during the neurodevelopmental period, as well as neurogenesis in the adult brain [209]. Moreover, neurotrophins have been linked to neurodegenerative and neuropsychiatric diseases such as Alzheimer’s disease (AD) [210] or bipolar disorder [211]. A successful example of this approach is adalimumab, a tumour necrosis factor-alpha (TNF-α) blocker, which was re-engineered for BBB penetration in a fusion protein with the human insulin receptor monoclonal antibody (HIRMAb). The fusion protein is known as HIRMAb-TNFR, which undergoes receptor-mediated transport across the BBB via the endogenous insulin receptor, thus inhibiting TNF-α by releasing a fused TNF inhibitor [212]. The effectiveness of the molecular Trojan Horse approach has been tested in animal models of stroke, cancer, Parkinson’s disease (PD), multiple sclerosis (MS), and AD, among others, as has been extensively reviewed by Pardridge [213].

#### 5.3.2. Nanotechnology Approaches

Nanotechnology is an innovative resource to deliver drugs within the brain; a wide range of pharmaceutical nano-carriers including liposomes, micelles, dendrimers, and some others have been developed. For instance, a wide range of nano-treatments were generated as approaches for AD with the potential to produce a beneficial impact for patients [214]. Pai et al. revealed that stabilized polyethylene glycol phospholipid nano-micelles were effective in mitigating Aβ deposition and neurotoxicity in an SHSY-5Y human cell model [214]. Moreover, antibodies can be potential therapies, too, as it was demonstrated using an adeno-associated virus to deliver antibodies against mutant SOD1 in ALS mouse model, revealing an attenuation of motor loss, a reduction of neural stress and alleviation of gliosis [215].

#### 5.3.3. Exosomes Therapies

Another remarkable potential therapy relates to exosomes, nano-vesicles secreted by the cells, acting as an intercellular communication and able to cross the BBB. Exosomes have been widely studied in disease since they are secreted into the extracellular space and may be useful diagnostically as well as in potential therapeutic approaches [216]. They participate in neurodegenerative diseases such as AD, ALS [217], HD [218], or CJD [219]. Kokubo et al. reported Aβ accumulation in exosomes within senile plaques in an AD mouse model [220]. In parallel, An et al. demonstrated that exosomes derived from human cerebrospinal fluid can reduce the action of Aβ in vivo [221].

Perets et al. reviewed the effects of the exosomes derived from bone marrow mesenchymal stem cells (MSCs) in different brain pathologies. They outlined an acute accumulation of exosomes in stroke, ASD, AD and PD, in contrast to a more diffuse pattern in healthy controls. The deposition of exosomes was found to be increased in disease-specific brain regions depending on the disease, for example, hippocampus in AD [222] or cerebellum in ASD [223]. With these findings, they proposed that exosomes could represent potential biomarkers or exploited to deliver therapeutic agents [224].

#### 5.3.4. Microbiome Therapies

A body of evidence has emerged outlining the relationship between the altered microbiome and pathological conditions. Because of this, the use of microorganisms and probiotics as potential therapies had been explored due to their influence on neuroinflammation in neurodegenerative conditions [225].

Mancuso et al. reviewed the relationship between AD and gut microbiota in preclinical and clinical studies and proposed the modulation of the gut microbiota as a potential therapy for the disease [226]. This was based on clinical studies such as the report from Tillish et al. which demonstrated that the consumption of fermented milk enriched with probiotics modulates the activity in specific regions of the brain in healthy women [227]. Moreover, Messaoudi et al. observed improved human behaviour with reduced anxiety, stress, negative mood, as well as a decreased level of urinary cortisol after 30 days of probiotics administration [228,229].

In neurodegenerative diseases, microbiota, probiotics and even antibiotics as targets for potential therapies is an emerging field [230,231]. Clinical studies have shown the beneficial effects of probiotics in MS, as Kouchaki et al. demonstrated a more favourable score in the disability status scale, inflammatory markers, mental health and lipid metabolism after a 12-week regimen of *Lactobacilli* and *Bifidobacterium* administered to patients [232].

The antibiotic doxycycline has been described to exhibit neuroprotective activity in animal models of PD [233], as well as reduced neurotoxic effects from microglia and astrocytes [234]. The main limitation of antibiotic administration is that it may affect non-target microorganisms with potential detrimental effects.

In parallel to neurodegenerative disease approaches, probiotics as a potential therapeutic strategy in neurodevelopmental pathologies had been explored, especially in relation to ASD. ASD patients have been reported to have an altered gastrointestinal barrier [235], and the use of probiotics may prevent ASD symptomatology, as it was reported by Hsiao et al. The authors demonstrated alteration in the maternal immune system, and with the administration of *Bacteroides fragilis*, the offspring developed correct gut permeability; *Bacteroides fragilis*, in turn, alters the microbial composition and reduces ASD-like behaviours including impairments of communication, anxiety and altered sensorimotor activities [236].

### 5.4. Stem Cell-Based Techniques

Lastly, stem cell-based therapies are changing scientist strategies toward the cure for neural disorders. In this context, BBB impairment can be used as an ally, as increased permeability can facilitate the migration of stem cells from the blood to the brain. Thus, BBB leakage and the pro-inflammatory environment caused by disease can be exploited for drug delivery or as coadjutants [237] for cell transplantation [238].

Embryonic stem cells (ESCs) are a great source for cell-based therapies, but, due to ethical concerns, adult stem cells are usually preferred for transplantation. There are three main types of adult-derived stem cell that are applied in neuroprotective therapeutic approaches: mesenchymal stem cells (MSCs), neural stem cells (NSCs) and induced-pluripotent stem cells (i-PSCs).

**MSCs** are the most widely used stem cells in brain-related treatments. Despite the migrating limitations enforced by the BBB in many situations such as TBI, MSCs can migrate across the endothelial cells via paracellular pathways through transiently formed inter-endothelial gaps [239]. Therefore, they can be used either for cell-mediated delivery of different elements, e.g., neuroprotective factors, or as a source for cellular regeneration. The applications of MSCs have been extensively employed in pre-clinical studies using various animal models as well as several clinical trials for PD, AD, ALS, MS, stroke [240] and ASD [241].

**NSCs** are widely found in foetal tissue and in a very limited region in adult brains, e.g., subventricular zone [242]. However, they can be generated in vitro using i-PSCs by passing the ethical issues derived from using foetal-derived tissue. NSC transplantation has been shown to improve AD features in animal models, e.g., memory deficit [243], by releasing neurotrophic factors such as BDNF [244] or participating in disaggregation of Aβ plaques [245].

Finally, since Takahashi and Yamanaka discovered the four transcriptional factors, i.e., Oct3/4, Sox2, c-Myc, and Klf4 that can induce pluripotency into adult cells [246,247], the use of **i-PSCs** for in vitro modeling has exponentially increased during the last decade [248]. Moreover, these cells can be used as therapeutic agents by differentiating them into the target cells altered in a disorder [249], as demonstrated in PD. In this pathology, the transplantation of dopaminergic progenitor cells into the putamen reported clinical changes gradually during the 18 to 24 months after implantation in the patient [250].

However, the safety and reliability of i-PSCs in the therapeutic arena is controversial and requires further efforts to overcome important key limitations, e.g., sporadic differentiation [251].

Overall, the delivery method is the main limitation for stem cell therapies. Direct transplantation at the site of injury is thought to be the more efficient, although this may cause injury at the site of injection [252]. Alternatively, intravenous injection has been reported to cause MSCs to be trapped in the lung vasculature [253]; nonetheless, they can migrate to other tissues, e.g., the brain, rescuing memory deficit and neuropathology in an AD mouse model [239]. Alternatively, stem cells may be delivered intranasally [254] and intrathecally [255], both feasible and accessible alternative routes by circumventing the BBB.

Notwithstanding, there is no consensus on either the adequate stem cell or the delivery route in the treatment of brain-related disorders.

## 6. Conclusions

The blood–brain barrier (BBB) is a unique structure consisting of different cell types such as brain endothelial cells, pericytes and astrocytes, thus constituting a functional entity, the neurovascular unit. The barrier’s establishment and formation take time in a multistep and overlapping process, where BBB formation or angiogenesis and BBB differentiation occur at prenatal age; maturation and maintenance of the newly formed BBB happen postnatally. A wide range of publications reporting BBB integrity loss in pathological conditions are reviewed in this manuscript, from Alzheimer’s disease (AD) to traumatic brain injury. A potential picture emerges indicating that BBB disruption might be not only a mechanism common to a variety of developmental and neurodegenerative conditions, but, in many cases, it might be a triggering event, leading to CNS dysfunction.

The correct function of the BBB rests on the interplay between many different cell types and the evolution of their cross-talk over time. Indeed, tuned expression of tight junctions (TJs), basement membrane components (BM) such as collagen IV or fibronectin, and proteins involved in the BBB maintenance and transport such as ATP-binding cassette (ABC) or excitatory amino-acid (EAAT) transporters are at the core of BBB functionality. As reviewed, many of the genetic causes linked to neurodegenerative diseases and many of the stressors that cause neurodevelopmental disorders trigger mechanisms such as neuroinflammation, oxidative stress and changes in neurotransmitter and ion homeostasis. The pathophysiology of these pathways has been associated with alterations in the function of astrocytes, which are, in fact, a common denominator in neuronal support and BBB function.

Indeed, although in neurodegenerative diseases such as AD, PD and ALS the focus has been specifically on neurons for many decades, astrocyte dysfunction in now well known. As we have moved away from the idea that astrocytes are simply reacting to neuronal death, and they might be affected directly by the same genetic and environmental factors that drive neuronal degeneration, we can hypothesize that these glial cells are intrinsically dysfunctional in disease and affect neurons and BBB function through the same pathways.

In the same context, dysfunction of brain microvascular endothelial cells, the main cell type forming the BBB, has been reported in the early stages of neurological disorders such as AD and neurovascular dementia. In fact, recent studies suggest an early BBB breakdown in patients, appearing as microbleeds. Therefore, brain endothelial cells must be considered as a potential therapeutic target, together with the other components of the NVU.

A remarkable new interest on microbiome modulation and its application as a therapeutic agent has arisen during the last decade. Therefore, the gut–brain axis involvement in health and disease progression is also a topic of interest in this review. Interestingly, microbiota may contribute to BBB modulation and disruption, triggering neuroinflammatory mechanisms. Autism spectrum disorders (ASD) are one of the most common pathologies where microbiome alteration has been reported. The concept that the microbiome may interfere with the BBB development leading to an alteration in the permeability of the barrier has raised many questions in the field and initiated new lines of research. The idea that BBB disruption during the prenatal period due to a maternal infection might predispose the person to neurological conditions later in life is one of the emerging new areas of investigation to understand the contribution of microbiota to BBB dysfunction.

In the last section, therapies commonly used and novel therapeutic approaches have been reviewed. Some classic approaches such as neurosurgery (widely applied in cancer); biology-based therapies including Trojan Horse technology, and chemistry-based therapies exemplified by halogenation were summarized. Furthermore, new trending treatments such as exosomes and nano-vesicles administration, microbiome modulation e.g., antibiotics, and stem-cell-based therapies including the mesenchymal stem cells (MSCs) were discussed. A great example of the Trojan Horse technology is adalimumab, a tumour necrosis factor-alpha blocker conjugated with a human insulin receptor antibody (HIRMAb-TNFR) able to penetrate the BBB. This drug has been tested in a variety of disorders, including AD and PD, due to the role of TNF-α in neuroinflammation, a common factor in several pathological conditions. The role of the microbiome in the maintenance of the BBB highlights the use of probiotics and antibiotics as potential therapies in some contexts such as ASD or PD.

In summary, several new techniques are being developed to allow therapeutic agents to cross the BBB and achieve innovative treatments for neurological disorders. However, the heterogeneity of brain pathologies means that scientists have a long way to go and more research is still needed.

The data reviewed here highlight the pathophysiological commonalities between neurodevelopmental and neurodegenerative disorders, including mechanisms such as hypoxia, neuroinflammation, physical injury and microbiota alterations. For this reason, research in both groups of disorders is necessary to understand whether there is a link between developmental and childhood disorders and the predisposition to develop neurodegenerative conditions later in life. We believe this is a compelling area of future research, where either chemistry-, biology- and or stem cell-based therapeutic approaches could be used for addressing a large variety of CNS pathologies.

## Figures and Tables

**Figure 1 ijms-23-15271-f001:**
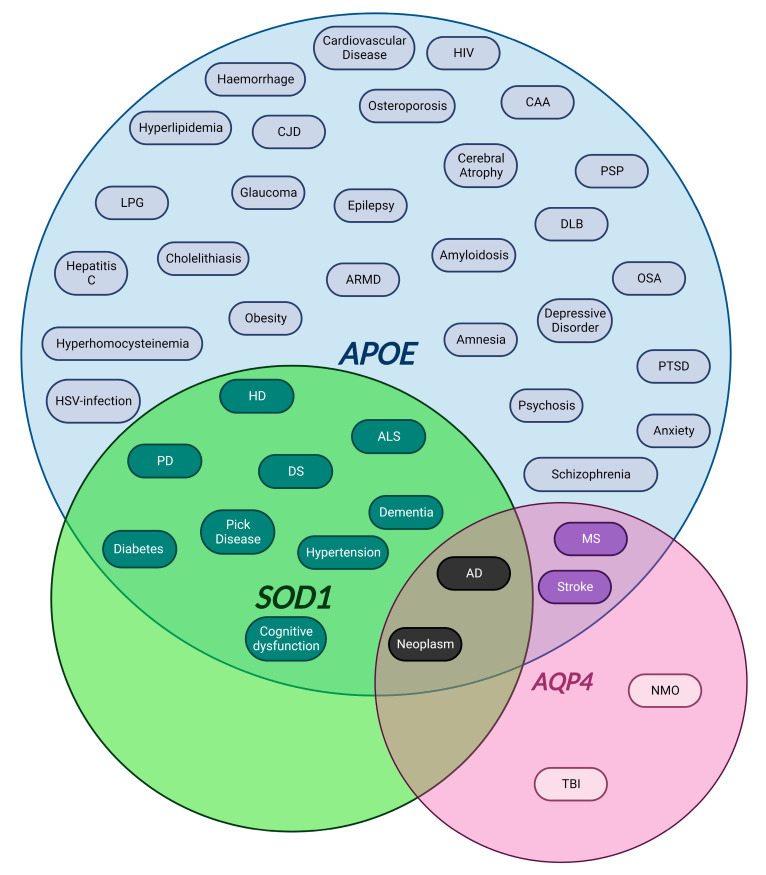
APOE, SOD1 and AQP4 genes are involved in multiple brain disorders and other pathologies. To create the Venn Diagram, diseases reported in ten or more research studies were selected. All the data related to APOE, SOD1, AQP4 and their variants associated with human diseases were explored in the platform DisGeNET. Acronyms used: Age-related Macular Degeneration (ARMD), Alzheimer’s Disease (AD), Amyotrophic Lateral Sclerosis (ALS), Congenital Aural Atresia (CAA), Creutzfeldt–Jakob disease (CJD), Down Syndrome (DS), Herpes Simplex Virus Infections (HSV-infection), Huntington Disease (HD), Traumatic Brain Injury (TBI), Lewy Body Disease (DLB), Lipoprotein Glomerulopathy (LPG), Multiple Sclerosis (MS), Neuromyelitis Optica (NMO), Parkinson’s Disease (PD), Post-Traumatic Stress Disorder (PTSD), Progressive Supranuclear Palsy (PSP), and Obstructive Sleep Apnoea (OSA).

**Figure 2 ijms-23-15271-f002:**
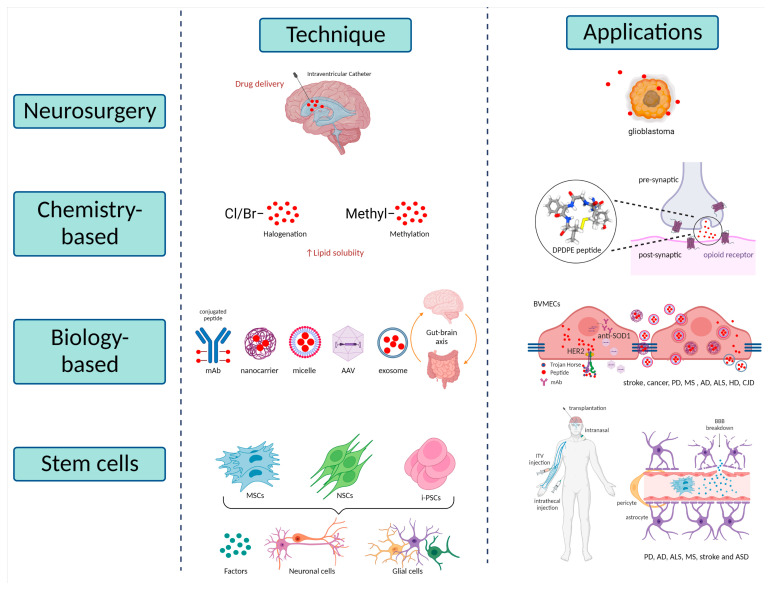
Main potential therapies and their most extended applications. The potential therapies described here are: neurosurgery [203,204], chemistry-based [202,205,206,207], biology-based [208,209,210,211,212,213,214,215,216,217,218,219,220,221,222,223,224,225,226,227,228,229,230,231,232,233,234,235,236] and stem cell-based [237,238,239,240,241,242,243,244,245,246,247,248,249,250,251,252,253,254,255] approaches. Due to the brain’s lack of a lymphatic system, the drugs administrated by neurosurgical techniques preferentially move along white matter tracts, thus are widely utilised in glioblastoma treatment. Chemistry-based therapies are made to increase drug solubility and targeting scope, such as lipophilicity modulation, halogenation or methylation. Chemical approaches and biology-based therapies, as Trojan Horse technology or exosomes administration are often combined to treat a wide range of neurological disorders. Stem cells can exert their therapeutic effects either through their neurotrophic capacity or through cell replacement. This figure has been designed as a summary of the Potential Therapies section; please refer to the text for further details. This figure was created using biorender.com.

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
