# Peer review of "Blood–Brain Barrier Disruption and Its Involvement in Neurodevelopmental and Neurodegenerative Disorders"

_ijms, 2022, doi:10.3390/ijms232315271_

Round 1
Reviewer 1 Report
González et submitted the manuscript, "Blood-Brain Barrier disruption and its involvement in neurodevelopmental and neurodegenerative disorders," written in a systematic way.
There are certain points that need to be addressed
1. There are various typographical errors in the paper, that require a major revision; please follow the journal guidelines.
2. Please improve Figure 2, improve the text embedded in the figure, and re-correct the figure caption.
3. Please elaborate on the "Trojan Horse technology" and provide more description.
4. Chemistry-based therapies need more description and add more technical details.
5. Blood-brain Barrier (BBB), please provide more description related to the molecular structural information, especially hydrophobicity at the cellular level.
There are various typographical errors throughout the manuscript, which require a thorough revision before considering for the current journal.
Reviewer 2 Report
The review entitled (Blood-Brain Barrier disruption and its involvement in neurodevelopmental and neurodegenerative disorders) submitted by Aragón-González et al., discusses BBB disruption as a common aspect in neurodegenerative diseases and other related disorders. In the review, conditions altering the BBB during the earliest and latest stages of life were discussed, revealing common factors involved. Also, potential therapies based on the BBB properties as molecular Trojan horses were reviewed, as well as innovative treatments including stem-cell therapies. Also, microflora modulation strategies were discussed. Future research directions were also highlighted. However, the review can not be accepted in its current form. A major revision is required as follows:
1. Resolution of Figures should be improved.
2. In the figure 2 caption, as well as in other sections, the reference format should be checked.
3. Plz check abbreviations and define them where they were first mentioned.
4. The number of references is too many. average of 200 is acceptable.
5. The role of APOE is not well discussed. please paraphrase this section to be easily understood.
6. In section 5, potential therapies, there is a problem of reference. ((Error! Reference source not found.))
7. The conclusion section and future directions should be expanded and discussed in detail to comprehensively summarize the point of view of the authors.
Good luck
Round 2
Reviewer 1 Report
authors revised the manuscript accordingly
Reviewer 2 Report
The authors well addressed my comments and The manuscript is ready to be published